# Ferrous Salt-Catalyzed Oxidative Alkenylation of Indoles: Facile Access to 3-Alkylideneindolin-2-Ones

Yunfei Tian [1,*], Luping Zheng [1], Ying Chen [1], Yufei Li [1], Mengna Wang [1], Weijun Fu [1] and Zejiang Li [2,*]

1 Key Laboratory of Fuction-Oriented Porous Materials of Henan Province, College of Chemistry and Chemical Engineering, Luoyang Normal University, Luoyang 471934, China
2 Key Laboratory of Medicinal Chemistry and Molecular Diagnosis of the Ministry of Education, College of Chemistry and Material Science, Key Laboratory of Chemical Biology of Hebei Province, Hebei University, Baoding 071002, China
* Correspondence: tianyunfly1120@163.com (Y.T.); lizejiang898@126.com (Z.L.)

**Abstract:** The direct oxidative alkenylation of indoles is achieved by ferrous salts under mild conditions, which provides one effective strategy for the synthesis of 3-alkylideneindolin-2-one in a single step. This reaction system features simple and readily available materials, mild conditions, and easy accessibility. The control experiments also demonstrate a radical pathway was involved in the reaction. Moreover, the method performs well on the gram-scale experiment, which indicates that this method enjoys a broad prospect in synthetic chemistry.

**Keywords:** ferrous salts; radical; oxidative alkenylation; 3-alkylideneindolin-2-ones

## 1. Introduction

Oxindoles are prevalent motifs widely present in natural products, pharmaceuticals (Figure 1), and functional materials [1–3]. Among such, 3-alkylideneindolin-2-ones have attracted much attention because of their valuable biological activity and synthetic applicability [4]. Recently, 3-alkylideneindolin-2-one-based solar cells and organic field-effect transistors (OFETs) were developed by the Zhang [5] and Liu groups [6]. Indeed, 3-alkylideneindolin-2-ones were also pivotal precursors to constructing naturally occurring alkaloids and drug candidates [7,8]. The desirable properties of 3-alkylideneindolin-2-ones make them popular within the chemical community.

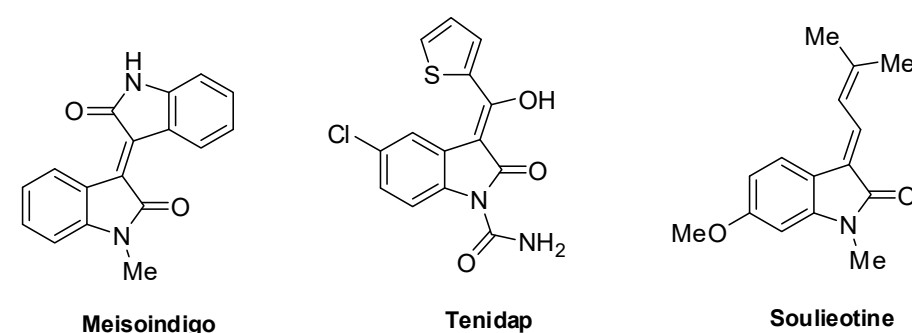

**Figure 1.** Representative examples of 3-alkylideneindolin-2-ones.

In addition, 3-alkylidene oxindoles were usually acquired by condensation reactions between oxindole and carbonyl compounds [9–11]; however, they need some harsh reaction conditions, such as stoichiometry metal catalysts and elevated temperature. On the other hand, diverse transition metal-catalyzed cyclization reactions have been frequently used for the synthesis of 3-alkylideneoxindoles. The above intramolecular cyclization systems are summarized as follows: the copper-catalyzed cyclization of β-keto amides [12],

palladium-promoted intramolecular aromatic C–H functionalization/C–C bond formation of alkenylamides [13] or 2-alkynylanilines [14–16], the carbonylation/cyclization of 2-alkynylanilines catalyzed by Pd [17], Rh [18], or Ni [19], and Pd/Rh initiated cyclization of 2-(alkynyl)aryl isocyanates with terminal alkynes/organoboronic acids [20–22]. Despite these apparent successes, the use of specially functionalized starting materials and noble transition metals limits their applications in scope. Therefore, some more straightforward, cost-effective, and environmentally benign approaches to access 3-alkylideneindolin-2-ones are being studied by chemists (Scheme 1).

**Scheme 1.** Synthesis methods of 3-alkylideneindolin-2-ones. **1-2** was two-steps processes for the preparation of 3-alkylideneindolin-2-ones.

As an alternative pathway, the direct synthesis of 3-alkylideneoxindole was also reported by many groups. For instance, Wan and coworkers reported 4-dimethylaminopyridine (DMAP)-promoted synthesis of 3-alkenyl-oxindoles by using isatins and acyl chlorides as starting materials [23]. Siddiki and coworkers developed a selective C3-alkenylation of oxindole with aldehydes [24]. Gnanaprakasam's group reported that Ru(II)-NHC promoted the synthesis of 3-(diphenylmethylene)indolin-2-one by using diaryl methanols and 2-oxindole as substrates [25]. Gopalaiah and coworkers finished iron-catalyzed direct access to (E)-3-alkylideneindolin-2-ones with oxindoles and benzylamines [26]. As we all know, the direct synthesis of 3-alkylideneoxindole was always involved in the oxindoles, which were synthesized by regioselective oxidation of indoles. In addition, the two-step protocol reduces the reaction efficiency and increases the operation complexity. In recent years, the difunctionalization of alkenes has attracted much attention in the fields of the construction of complex molecules [27–30]. Liu and coworkers reported the radical oxidative fluoroalkylfluorosulfonylation of unactivated alkenes [31]. Xi and coworkers developed photoredox-catalyzed direct keto-difluoroacetylation of styrenes with (fluorosulfonyl)difluoroacetate and dimethyl sulfoxide [32]. Li and coworkers realized radical-mediated alkoxypolyhaloalkylation of styrenes with polychloroalkanes and alcohols [33]. Our group also developed iron-mediated azidomethylation or azidotrideuteromethylation of active alkenes with azidotrimethylsilane and dimethyl sulfoxide [34]. Consequently, it is immensely valuable to afford 3-alkylideneindolin-2-ones by the direct difunctionalization of indoles. As our continuing interests in the preparation of heterocycles [35–37], we described an iron(II) salt-catalyzed oxidative alkenylation of indoles with carbonyl compounds to obtain 3-alkylideneindolin-2-ones in a mild condition.

## 2. Results

Initially, we began our investigation by exploring the reaction of 1-methylindole with acetone. To our delight, the desired product **1** was isolated in a 26% yield when the reaction was on treatment with 5 mol % FeCl$_2$ as a catalyst and hydrogen peroxide (H$_2$O$_2$) as an oxidant at 25 °C (Table 1, entry 1). By screening a series of solvents, we found acetone/H$_2$O (2.5/1) was the optimum cosolvent, which produced product **1** in a 45% yield (entries 1–3). Varying the equivalent of oxidants to 2 equiv., the yield of the desired product was increased to 51% (entries 4–7). However, other oxidants, such as DTBP, TBPA, and TBHP (in water), were not effective in this transformation (entries 8–10). Notably, after a variety of further optimization of catalysts, we found FeCl$_2$ was best for this oxidative alkenylation reaction (entries 11–18). Finally, via the variation of reaction temperature, product **1** could be isolated in a 72% yield at 45 °C (entries 19–21).

**Table 1.** Optimization of reaction conditions [a].

| Entry | Solvent (mL) | Catalyst (mol %) | Oxidant (equiv.) | T/°C | Yield [b] (%) |
|---|---|---|---|---|---|
| 1 | 5/1 (3.5) | FeCl$_2$ (5) | H$_2$O$_2$ (3) | 25 | 26 |
| 2 | 2.5/1 (3.5) | FeCl$_2$ (5) | H$_2$O$_2$ (3) | 25 | 45 |
| 3 | 1/1 (3.5) | FeCl$_2$ (5) | H$_2$O$_2$ (3) | 25 | 19 |
| 4 | 2.5/1 (3.5) | FeCl$_2$ (5) | H$_2$O$_2$ (1) | 25 | 28 |
| 5 | 2.5/1 (3.5) | FeCl$_2$ (5) | H$_2$O$_2$ (2) | 25 | 51 |
| 6 | 2.5/1 (3.5) | FeCl$_2$ (5) | H$_2$O$_2$ (5) | 25 | 39 |
| 7 | 2.5/1 (3.5) | FeCl$_2$ (5) | H$_2$O$_2$ (7) | 25 | 22 |
| 8 | 2.5/1 (3.5) | FeCl$_2$ (5) | DTBP (2) | 25 | 7 |
| 9 | 2.5/1 (3.5) | FeCl$_2$ (5) | TBPA (2) | 25 | 8 |
| 10 | 2.5/1 (3.5) | FeCl$_2$ (5) | TBHP(in H$_2$O) (2) | 25 | 12 |
| 11 | 2.5/1 (3.5) | Fe(SO$_4$)$_2$·7H$_2$O (5) | H$_2$O$_2$ (2) | 25 | 37 |
| 12 | 2.5/1 (3.5) | FeCl$_3$ (5) | H$_2$O$_2$ (2) | 25 | 17 |
| 13 | 2.5/1 (3.5) | CuCl$_2$ (5) | H$_2$O$_2$ (2) | 25 | 24 |
| 14 | 2.5/1 (3.5) | ZnBr$_2$ (5) | H$_2$O$_2$ (2) | 25 | NR |
| 15 | 2.5/1 (3.5) | FeCl$_2$ (0) | H$_2$O$_2$ (2) | 25 | trace |
| 16 | 2.5/1 (3.5) | FeCl$_2$ (2) | H$_2$O$_2$ (2) | 25 | 29 |
| 17 | 2.5/1 (3.5) | FeCl$_2$ (10) | H$_2$O$_2$ (2) | 25 | 45 |
| 18 | 2.5/1 (3.5) | FeCl$_2$ (20) | H$_2$O$_2$ (2) | 25 | 37 |
| 19 | 2.5/1 (3.5) | FeCl$_2$ (5) | H$_2$O$_2$ (2) | 35 | 65 |
| 20 | 2.5/1 (3.5) | FeCl$_2$ (5) | H$_2$O$_2$ (2) | 45 | 72 |
| 21 | 2.5/1 (3.5) | FeCl$_2$ (5) | H$_2$O$_2$ (2) | 55 | 42 |

[a] Reaction conditions: indole (1 equiv., 0.2 mmol), hydrogen peroxide (30%, 2 equiv., 0.4 mmol), ferrous chloride (5 mol %, 0.01 mmol), acetone (2.5 mL), H$_2$O (1 mL), 45 °C, and 14 h. [b] Isolated yields.

With the optimized conditions in hand, we went on to examine the substrate scopes of the reaction. As expected, various indoles underwent the reaction smoothly to afford the corresponding 3-alkylideneoxindoles with moderate to good yields (Scheme 2). Some sensitive functional groups (−F, −Cl, −Br, −I) were compatible with the optimal conditions, which showed the further synthetic potential of the method (**2–5**). Next, we investigated the electronic effect of the substituents of the substrates. The electron-donating substituent-revised indoles (such as 5-Me, 6-Me, 7-Me, and 5-OBn) gave the corresponding 3-alkylideneoxindoles in 66–79% yields (**6–9**); however, the electron-withdrawing groups containing substrates (-CO$_2$Et, -COCH$_3$, -NO$_2$, and -CN) failed to offer the final products under the standard conditions (see ESI). Some disubstituted indoles, such as 6-chloro-5-

fluoro-1-methylindole, did not work well in the reaction system (**10**). Meanwhile, the effect of various *N*-protecting groups of the starting materials was also investigated. Delightfully, all of the protected groups were well tolerant in the transformation, which resulted in the corresponding products in 60–80% yields (**11–15**). Notably, the reactive functional groups, such as ally and 2-hydroxyethyl, were also compatible with this system, and products **16–17** could be obtained in high yields. In addition, oxidative condensation of some unprotected *N*- moiety of indoles was also achieved with 30–40% yields (**18–21**). Additionally, the structure of product **21** was further confirmed by X-ray crystallography. Finally, this transformation could also proceed smoothly by using acetone-$d_6$ as the solvent, and the corresponding products **22** and **23** were isolated in 78–83% yields, respectively.

**Scheme 2.** Scope of various indoles. [a] Reaction conditions: indole (1 equiv., 0.2 mmol), ferrous chloride (5 mol %, 0.01 mmol), hydrogen peroxide (30%, 2 equiv., 0.4 mmol), acetone (2.5 mL), $H_2O$ (1 mL), 45 °C, 14 h, and isolated yields. [b] 35 °C.

Next, some control experiments were performed to examine the application value of this oxidative alkenylation reaction. First, the desired product **1** was isolated in a 75% yield when the system scaled up to 10 mmol (Scheme 3(1a)). Meanwhile, we also obtained the desired products with a 70% yield when the amount of catalyst was reduced to 3% (Scheme 3(1b)). Subsequently, some mechanistic studies were explored to verify the reaction process. After the addition of a radical inhibitor, such as BHT or TEMPO, the reaction was inhibited significantly (Scheme 3(2)). These results indicate that this reaction may involve a radical pathway.

**Scheme 3.** (**1**) Gram-scale experiments: (**1a**) standard conditions; (**1b**) 3 mL % ferrous chloride. (**2**) control experiments.

On the basis of the above results and literature precedent, a plausible mechanism pathway for the preparation of 3-alkylideneindolin-2-one is illustrated as shown in Scheme 4. First, with the assistance of Fe(II) salts, $H_2O_2$ involves a heterolysis reaction and offers a hydroxyl radical and hydroxyl anion. Then, the hydroxyl radical reacts with *N*-methylindole to form the radical **A**, which is oxidized by Fe(III) to form the carbocation **B**. Intermediate **B** loses a proton to yield intermediate **C**, which is easily converted into isomer **D**. Subsequently, the isomer **D** loses a proton to gain the anionic **E**. Finally, intermediate **E** reacts with acetone and proceeds with addition/elimination to access the desired product **1**.

**Scheme 4.** Plausible mechanism. (**A**–**F**) were possible intermediates to demonstrate the reaction mechanism. (**1**) was the desired product.

## 3. Materials and Methods

### 3.1. Materials

[1]H and [13]C NMR spectra were recorded on a Bruker advance III 500 or 400 spectrometer in $CDCl_3$ with TMS as the internal standard. High-resolution mass spectral analysis (HRMS(TOF)) data were measured on a Bruker Apex II. All products were identified by [1]H and [13]C NMR. The starting materials were purchased from Energy, J&K Chemicals, or

Aldrich and used without further purification. The conversion was monitored by thin-layer chromatography (TLC). Flash column chromatography was performed over silica gel (200–300 mesh).

*3.2. Methods*

Indole (1 equiv., 0.2 mmol) and ferrous chloride (5 mol %, 0.01 mmol) were added to a 20 mL test tube with a magnetic stir bar. Then, acetone (2.5 mL), $H_2O$ (1 mL), and $H_2O_2$ (30%, 2 equiv., 0.4 mmol) were slowly added to the mixture, respectively. The resulting reaction mixture was allowed to stir at 45 °C or 35 °C (oil bath) for 14 h. After cooling to room temperature, the reaction mixture was diluted with saturated brine (10 mL) and extracted with EtOAc. The combined organic layers were dried over $Na_2SO_4$ and concentrated in vacuo. The crude product was purified by flash chromatography using PE/EA as eluent to afford the desired products (**1–23**). (See Supplementary Materials).

*1-methyl-3-(propan-2-ylidene)indolin-2-one* (**1**). A colorless liquid after purification by flash column chromatography (petroleum ether/ethyl acetate = 40/1), 26.9 mg, with yield of 72%. $^1$H NMR (500 MHz, $CDCl_3$): $\delta$ 7.43 (d, *J* = 7.5 Hz, 1H), 7.20 (t, *J* = 7.5 Hz, 1H), 6.96 (t, *J* = 7.5 Hz, 1H), 6.72 (d, *J* = 7.5 Hz, 1H), 3.15 (s, 3H), 2.56 (s, 3H), and 2.28 (s, 3H). $^{13}$C NMR (125 Hz, $CDCl_3$): $\delta$ 167.5, 154.4, 141.7, 127.2, 123.3, 123.0, 122.4, 121.3, 107.2, 25.3, 24.9, and 22.9. HRMS (ESI, *m/z*): Calculated for $C_{12}H_{13}NO$ [M + H]$^+$ 188.1070, found 188.1071.

*5-fluoro-1-methyl-3-(propan-2-ylidene)indolin-2-one* (**2**). A colorless liquid after purification by flash column chromatography (petroleum ether/ethyl acetate = 40/1), 23.4 mg, with yield of 57%. $^1$H NMR (400 MHz, $CDCl_3$): $\delta$ 7.27 (dd, *J* = 9.2, 2.4 Hz, 1H), 6.96 (td, *J* = 8.8, 2.4 Hz, 1H), 6.71 (dd, *J* = 8.4, 4.4 Hz, 1H), 3.23 (s, 3H), 2.64 (s, 3H), and 2.36 (s, 3H). $^{13}$C{$^1$H} NMR (100 MHz, $CDCl_3$): $\delta$ 167.7, 158.6 (d, *J* = 236.8 Hz), 156.5, 138.0, 124.4 (d, *J* = 8.8 Hz), 122.6 (d, *J* = 2.9 Hz), 113.4 (d, *J* = 23.6 Hz), 111.2 (d, *J* = 26.3 Hz), 107.5 (d, *J* = 8.4 Hz), 25.7, 25.1, and 23.3. HRMS (ESI, *m/z*): Calculated for $C_{12}H_{12}FNO$ [M + H]$^+$ 206.0976, found 206.0979.

*5-chloro-1-methyl-3-(propan-2-ylidene)indolin-2-one* (**3**). A colorless liquid after purification by flash column chromatography (petroleum ether/ethyl acetate = 40/1), 29.6 mg, with yield of 67%. $^1$H NMR (400 MHz, $CDCl_3$): $\delta$ 7.49 (d, *J* = 2.0 Hz, 1H), 7.22 (dd, *J* = 8.4, 2.0 Hz, 1H), 6.73 (d, *J* = 8.4 Hz, 1H), 3.23 (s, 3H), 2.64 (s, 3H), and 2.38 (s, 3H). $^{13}$C{$^1$H} NMR (100 MHz, $CDCl_3$): $\delta$ 167.5, 156.8, 140.5, 127.1, 126.8, 124.8, 123.5, 122.1, 108.2, 25.7, 25.3, and 23.3. HRMS (ESI, *m/z*): Calculated for $C_{12}H_{12}ClNO$ [M + H]$^+$ 222.0680, found 222.0681.

*5-bromo-1-methyl-3-(propan-2-ylidene)indolin-2-one* (**4**). A colorless liquid after purification by flash column chromatography (petroleum ether/ethyl acetate = 40/1), 33.0 mg, with yield of 62%. $^1$H NMR (500 MHz, $CDCl_3$): $\delta$ 7.60 (s, 1H), 7.34 (d, *J* = 8.0 Hz, 1H), 6.66 (d, *J* = 8.0 Hz, 1H), 3.20 (s, 3H), 2.62 (s, 3H), and 2.35 (s, 3H). $^{13}$C{$^1$H} NMR (125 MHz, $CDCl_3$): $\delta$167.3, 156.8, 140.9, 130.0, 126.1, 125.3, 121.9, 114.2, 108.7, 25.6, 25.3, and 23.3. HRMS (ESI, *m/z*): Calculated for $C_{12}H_{12}BrNO$ [M + H]$^+$ 266.0175, found 266.0178.

*5-iodo-1-methyl-3-(propan-2-ylidene)indolin-2-one* (**5**). A colorless liquid after purification by flash column chromatography (petroleum ether/ethyl acetate = 40/1), 40.7 mg, yield 65%. $^1$H NMR (500 MHz, $CDCl_3$): $\delta$ 7.79 (s, 1H), 7.55 (dd, *J* = 8.0, 1.0 Hz, 1H), 6.59 (d, *J* = 8.0 Hz, 1H), 3.21 (s, 3H), 2.62 (s, 3H), and 2.36 (s, 3H). $^{13}$C{$^1$H} NMR (125 MHz, $CDCl_3$): $\delta$ 167.1, 156.8, 141.5, 136.0, 131.8, 125.8, 121.7, 109.4, 84.1, 25.6, 25.4, and 23.3. HRMS (ESI, *m/z*): $C_{12}H_{12}INO$ [M + H]$^+$ 314.0036, found 314.0037.

*1,5-dimethyl-3-(propan-2-ylidene)indolin-2-one* (**6**). A colorless liquid after purification by flash column chromatography (petroleum ether/ethyl acetate = 40/1), 29.3 mg, with yield of 73%. $^1$H NMR (400 MHz, $CDCl_3$): $\delta$ 7.34 (s, 1H), 7.05 (d, *J* = 7.6 Hz, 1H), 6.70 (d, *J* = 7.6 Hz, 1H), 3.22 (s, 3H), 2.62 (s, 3H), 2.37 (s, 3H), and 2.36 (s, 3H). $^{13}$C{$^1$H} NMR (100 MHz, $CDCl_3$): $\delta$ 167.9, 154.3, 139.8, 130.8, 127.7, 124.2, 123.6, 122.8, 107.1, 25.6, 25.2, 23.1, and 21.4. HRMS (ESI, *m/z*): Calculated for $C_{13}H_{15}NO$ [M + H]$^+$ 202.1226, found 202.1227.

*1,6-dimethyl-3-(propan-2-ylidene)indolin-2-one* (**7**). A colorless liquid after purification by flash column chromatography (petroleum ether/ethyl acetate = 40/1), 26.5 mg, with

yield of 66%. $^1$H NMR (400 MHz, CDCl$_3$): δ 7.40 (d, *J* = 8.0 Hz, 1H), 6.83 (d, *J* = 7.6 Hz, 1H), 6.64 (s, 1H), 3.22 (s, 3H), 2.60 (s, 3H), 2.39 (s, 3H), and 2.34 (s, 3H). $^{13}$C{$^1$H} NMR (100 MHz, CDCl$_3$): δ 168.1, 153.2, 142.1, 137.7, 123.1, 122.6, 122.1, 121.0, 108.4, 25.5, 25.1, 22.9, and 21.8. HRMS (ESI, *m/z*): Calculated for C$_{13}$H$_{15}$NO [M + H]$^+$ 202.1226, found 202.1230.

*1,7-dimethyl-3-(propan-2-ylidene)indolin-2-one* (**8**). A colorless liquid after purification by flash column chromatography (petroleum ether/ethyl acetate = 40/1), 27.3 mg, with yield of 68%. $^1$H NMR (400 MHz, CDCl$_3$): δ 7.40 (d, *J* = 7.5 Hz, 1H), 6.98 (d, *J* = 7.6 Hz, 1H), 6.91 (t, *J* = 7.6 Hz, 1H), 3.54 (s, 3H), 2.63 (s, 3H), 2.59 (s, 3H), and 2.36 (s, 3H). $^{13}$C{$^1$H} NMR (100 MHz, CDCl$_3$): δ 168.5, 154.3, 139.9, 131.3, 124.2, 122.5, 121.4, 121.3, 119.0, 29.0, 25.4, 23.6, and 19.5. HRMS (ESI, *m/z*): Calculated for C$_{13}$H$_{15}$NO [M + H]$^+$ 202.1226, found 202.1228.

*5-(benzyloxy)-1-methyl-3-(propan-2-ylidene)indolin-2-one* (**9**). A colorless liquid after purification by flash column chromatography (petroleum ether/ethyl acetate = 20/1), 46.3 mg, with yield of 79%. $^1$H NMR (500 MHz, CDCl$_3$): δ 7.45 (d, *J* = 7.5 Hz, 2H), 7.39 (t, *J* = 7.5 Hz, 2H), 7.33 (t, *J* = 7.5 Hz, 1H), 7.22 (d, *J* = 2.0 Hz, 1H), 6.86 (dd, *J* = 8.5, 2.5 Hz, 1H), 6.69 (d, *J* = 8.0 Hz, 1H), 5.06 (s, 2H), 3.21 (s, 3H), 2.62 (s, 3H), and 2.32 (s, 3H). $^{13}$C{$^1$H} NMR (125 MHz, CDCl$_3$): δ 167.8, 155.1, 154.3, 137.2, 136.4, 128.6, 128.0, 127.6, 124.5, 123.0, 112.8, 112.7, 107.3, 71.1, 29.7, 25.7, 25.1, and 23.1. HRMS (ESI, *m/z*): Calculated for C$_{19}$H$_{19}$NO$_2$ [M + H]$^+$ 294.1488, found 294.1490.

*6-chloro-5-fluoro-1-methyl-3-(propan-2-ylidene)indolin-2-one* (**10**). A colorless liquid after purification by flash column chromatography (petroleum ether/ethyl acetate = 40/1), 13.8 mg, with yield of 29%. $^1$H NMR (400 MHz, CDCl$_3$): δ 7.32 (d, *J* = 10.0 Hz, 1H), 6.78 (d, *J* = 6.0 Hz, 1H), 3.21 (s, 3H), 2.62 (s, 3H), and 2.34 (s, 3H). $^{13}$C{$^1$H} NMR (100 MHz, CDCl$_3$): δ 167.4, 157.1, 153.7 (d, *J* = 239.7 Hz), 138.6, 122.8 (d, *J* = 7.8 Hz), 121.9 (d, *J* = 2.5 Hz), 119.2 (d, *J* = 19.7 Hz), 112.0 (d, *J* = 25.9 Hz), 108.6, 25.8, 25.1, and 23.3. HRMS (ESI, *m/z*): Calculated for C$_{12}$H$_{11}$ClFNO [M + H]$^+$ 240.0586, found 240.0589.

*1-ethyl-3-(propan-2-ylidene)indolin-2-one* (**11**). A colorless liquid after purification by flash column chromatography (petroleum ether/ethyl acetate = 40/1), 28.1 mg, with yield of 70%. $^1$H NMR (400 MHz, CDCl$_3$): δ 7.54 (d, *J* = 7.6 Hz, 1H), 7.24 (t, *J* = 8.0 Hz, 1H), 7.02 (td, *J* = 7.6, 1.2 Hz, 1H), 6.84 (d, *J* = 7.6 Hz, 1H), 3.81 (q, *J* = 7.2 Hz, 2H), 2.63 (s, 3H), 2.38 (s, 3H), and 1.26 (t, *J* = 7.2 Hz, 3H). $^{13}$C{$^1$H} NMR (100 MHz, CDCl$_3$): δ 167.4, 154.6, 141.0, 127.4, 123.8, 123.5, 122.7, 121.3, 107.5, 34.0, 25.2, 23.1, and 12.8. HRMS (ESI, *m/z*): Calculated for C$_{13}$H$_{15}$NO [M + H]$^+$ 202.1226, found 202.1229.

*1-isopropyl-3-(propan-2-ylidene)indolin-2-one* (**12**). A colorless liquid after purification by flash column chromatography (petroleum ether/ethyl acetate = 40/1), 29.7 mg, with yield of 69%. $^1$H NMR (500 MHz, CDCl$_3$): δ 7.55 (d, *J* = 8.0 Hz, 1H), 7.21 (t, *J* = 8.0 Hz, 1H), 7.02–6.99 (m, 1H), 4.79–4.71 (m, 1H), 2.63 (s, 3H), 2.37 (s, 3H), and 1.49 (d, *J* = 7.0 Hz, 6H). $^{13}$C{$^1$H} NMR (125 MHz, CDCl$_3$): δ 167.4, 154.3, 140.6, 127.1, 124.1, 123.6, 122.8, 120.9, 109.1, 42.9, 25.3, 23.2, and 19.5. HRMS (ESI, *m/z*): Calculated for C$_{14}$H$_{17}$NO [M + H]$^+$ 216.1383, found 216.1381.

*1-butyl-3-(propan-2-ylidene)indolin-2-one* (**13**). A colorless liquid after purification by flash column chromatography (petroleum ether/ethyl acetate = 40/1), 34.3 mg, with yield of 75%. $^1$H NMR (500 MHz, CDCl$_3$): δ 7.53 (d, *J* = 7.5 Hz, 1H), 7.22 (td, *J* = 7.5, 1.0 Hz, 1H), 7.01 (td, *J* = 7.5, 1.0 Hz, 1H), 6.83 (dd, *J* = 8.0, 1.0 Hz, 1H), 3.74 (t, *J* = 7.5 Hz, 2H), 2.63 (s, 3H), 2.37 (s, 3H), 1.68–1.61 (m, 2H), 1.43–1.36 (m, 2H), and 0.95 (t, *J* = 7.5 Hz, 3H). $^{13}$C{$^1$H} NMR (125 MHz, CDCl$_3$): δ 167.70, 154.53, 141.53, 127.40, 123.76, 123.46, 122.74, 121.30, 107.71, 39.24, 29.80, 25.21, 23.13, 20.33, and 13.81. HRMS (ESI, *m/z*): Calculated for C$_{15}$H$_{19}$NO [M + H]$^+$ 230.1539, found 230.1540.

*1-benzyl-3-(propan-2-ylidene)indolin-2-one* (**14**). A colorless liquid after purification by flash column chromatography (petroleum ether/ethyl acetate = 40/1), 42.1 mg, with yield of 80%. $^1$H NMR (500 MHz, CDCl$_3$): δ 7.55 (d, *J* = 7.5 Hz, 1H), 7.30 (d, *J* = 4.5 Hz, 4H), 7.25–7.22 (m, 1H), 7.14 (t, *J* = 7.5 Hz, 1H), 7.01 (t, *J* = 7.5 Hz, 1H), 6.73 (d, *J* = 8.0 Hz, 1H), 4.98 (s, 2H), 2.68 (s, 3H), and 2.41 (s, 3H). $^{13}$C{$^1$H} NMR (125 MHz, CDCl$_3$): δ 167.8, 155.3,

141.2, 136.5, 128.7, 127.4, 127.3, 127.2, 123.7, 123.4, 122.5, 121.6, 108.5, 43.2, 29.7, 25.3, and 23.3. HRMS (ESI, *m/z*): Calculated for $C_{18}H_{17}NO$ [M + H]$^+$ 264.1383, found 264.1387.

*1-(propan-2-ylidene)-5,6-dihydro-1H-pyrrolo [3,2,1-ij]quinolin-2(4H)-one* (**15**). A colorless liquid after purification by flash column chromatography (petroleum ether/ethyl acetate = 40/1), 25.6 mg, with yield of 60%. $^1$H NMR (500 MHz, CDCl$_3$): δ 7.33 (d, *J* = 7.5 Hz, 1H), 6.99 (d, *J* = 8.0 Hz, 1H), 6.92 (t, *J* = 7.5 Hz, 1H), 3.76–3.73 (m, 2H), 2.79–2.77 (m, 2H), 2.61 (s, 3H), 2.35 (s, 3H), and 2.00 (dt, *J* = 12.0, 6.0 Hz, 2H). $^{13}$C{$^1$H} NMR (125 MHz, CDCl$_3$): δ 166.9, 154.5, 137.9, 126.3, 123.7, 122.1, 121.2, 121.0, 119.3, 38.2, 24.9, 24.8, 22.6, and 21.1. HRMS (ESI, *m/z*): Calculated for $C_{14}H_{15}NO$ [M + H]$^+$ 214.1226, found 214.1230.

*1-allyl-3-(propan-2-ylidene)indolin-2-one* (**16**). A colorless liquid after purification by flash column chromatography (petroleum ether/ethyl acetate = 40/1), 28.9 mg, with yield of 68%. $^1$H NMR (500 MHz, CDCl$_3$): δ 7.55 (d, *J* = 7.5 Hz, 1H), 7.21 (td, *J* = 7.5, 1.0 Hz, 1H), 7.03 (td, *J* = 7.5, 1.0 Hz, 1H), 6.82 (dd, *J* = 8.0, 1.0 Hz, 1H), 5.89–5.82 (m, 1H), 5.21–5.17 (m, 2H), 4.40 (dt, *J* = 5.0, 2.0 Hz, 2H), 2.64 (s, 3H), and 2.39 (s, 3H). $^{13}$C{$^1$H} NMR (125 MHz, CDCl$_3$): δ 167.4, 155.0, 141.2, 132.0, 127.4, 123.7, 123.4, 122.5, 121.52, 117.0, 108.3, 41.8, 25.2, and 23.2. HRMS (ESI, *m/z*): Calculated for $C_{14}H_{15}NO$ [M + H]$^+$ 214.1226, found 214.1229.

*1-(2-hydroxyethyl)-3-(propan-2-ylidene)indolin-2-one* (**17**). A colorless liquid after purification by flash column chromatography (petroleum ether/ethyl acetate = 10/1), 24.3 mg, with yield of 56%. $^1$H NMR (500 MHz, CDCl$_3$): δ 7.53 (d, *J* = 7.5 Hz, 1H), 7.23 (t, *J* = 8.0 Hz, 1H), 7.04 (t, *J* = 7.5 Hz, 1H), 6.90 (d, *J* = 7.5 Hz, 1H), 3.94–3.91 (m, 4H), 2.61 (s, 3H), and 2.38 (s, 3H). $^{13}$C{$^1$H} NMR (125 MHz, CDCl$_3$): δ 169.0, 155.8, 141.3, 127.5, 123.7, 123.5, 122.4, 121.8, 107.8, 61.3, 42.8, 25.3, and 23.3. HRMS (ESI, *m/z*): Calculated for $C_{13}H_{15}NO_2$ [M + H]$^+$ 218.1175, found 218.1176.

*3-(propan-2-ylidene)indolin-2-one* (**18**). A yellowish solid after purification by flash column chromatography (petroleum ether/ethyl acetate = 5/1), mp 187–188 °C, 11.4 mg, with yield of 33%. $^1$H NMR (500 MHz, CDCl$_3$): δ 8.45 (s, 1H), 7.52 (d, *J* = 7.6 Hz, 1H), 7.18 (t, *J* = 7.6 Hz, 1H), 7.05–6.98 (td, *J* = 7.6, 0.8 Hz, 1H), 6.87 (d, *J* = 7.6 Hz, 1H), 2.62 (s, 3H), and 2.38 (s, 3H). $^{13}$C{$^1$H} NMR (125 MHz, CDCl$_3$): δ 169.7, 155.6, 139.3, 127.5, 124.3, 123.7, 123.0, 121.5, 109.3, 25.2, and 23.1. HRMS (ESI, *m/z*): Calculated for $C_{11}H_{11}NO$ [M + H]$^+$ 174.0913, found 174.0917.

*7-methyl-3-(propan-2-ylidene)indolin-2-one* (**19**). A yellowish solid after purification by flash column chromatography (petroleum ether/ethyl acetate = 5/1), mp 201–202 °C, 13.1 mg, with yield of 35%. $^1$H NMR (500 MHz, CDCl$_3$): δ 8.80 (s, 1H), 7.32 (s, 1H), 6.99 (d, *J* = 8.0 Hz, 1H), 6.77 (d, *J* = 7.5 Hz, 1H), 2.61 (s, 3H), 2.37 (s, 3H), and 2.35 (s, 3H). $^{13}$C{$^1$H} NMR (125 MHz, CDCl$_3$): δ 155.1, 137.2, 130.6, 129.8, 127.9, 124.4, 123.3, 109.1, 25.2, 23.1, and 21.4. HRMS (ESI, *m/z*): Calculated for $C_{12}H_{13}NO$ [M + H]$^+$ 188.1070, found 188.1072.

*5-methoxy-3-(propan-2-ylidene)indolin-2-one* (**20**). A yellowish solid after purification by flash column chromatography (petroleum ether/ethyl acetate = 5/1), mp 199–200 °C, 16.2 mg, with yield of 40%. $^1$H NMR (500 MHz, CDCl$_3$): δ 8.44 (s, 1H), 7.12 (d, *J* = 1.5 Hz, 1H), 6.79–6.73 (m, 2H), 3.80 (s, 3H), 2.62 (s, 3H), and 2.36 (s, 3H). $^{13}$C{$^1$H} NMR (125 MHz, CDCl$_3$): δ 169.8, 155.8, 155.0, 133.3, 125.4, 123.4, 111.7, 111.6, 109.2, 56.0, 25.2, and 23.2. HRMS (ESI, *m/z*): Calculated for $C_{12}H_{13}NO_2$ [M + H]$^+$ 204.1019, found 204.1021.

*5-fluoro-3-(propan-2-ylidene)indolin-2-one* (**21**). A yellowish solid after purification by flash column chromatography (petroleum ether/ethyl acetate = 5/1), mp 217–218 °C, 11.5 mg, with yield of 30%. $^1$H NMR (400 MHz, CDCl$_3$): δ 8.53 (s, 1H), 7.24 (dd, *J* = 9.6, 2.0 Hz, 1H), 6.90 (td, *J* = 8.8, 2.4 Hz, 1H), 6.79 (dd, *J* = 8.4, 4.8 Hz, 1H), 2.63 (s, 3H), and 2.36 (s, 3H). $^{13}$C{$^1$H} NMR (100 MHz, CDCl$_3$): δ 169.7, 158.5 (d, *J* = 236.9 Hz), 157.4, 138.1, 135.3 (d, *J* = 1.5 Hz), 125.2 (d, *J* = 8.7 Hz), 113.7 (d, *J* = 23.8 Hz), 111.3 (d, *J* = 26.2 Hz), 109.5 (d, *J* = 8.4 Hz), 25.2, and 23.3. HRMS (ESI, *m/z*): Calculated for $C_{11}H_{10}FNO$ [M + H]$^+$ 192.0819, found 192.0820.

(**22**). A colorless liquid after purification by flash column chromatography (petroleum ether/ethyl acetate = 40/1), 28.5 mg, with yield of 78%. $^1$H NMR (400 MHz, CDCl$_3$): δ 7.34 (s, 1H), 7.05 (d, *J* = 8.0 Hz, 1H), 6.70 (d, *J* = 8.0 Hz, 1H), 3.21 (s, 3H), and 2.36 (s, 3H). $^{13}$C{$^1$H} NMR (100 MHz, CDCl$_3$): δ 167.9, 154.1, 139.8, 130.7, 127.7, 124.2, 123.6, 122.9, 107.1,

25.6, 24.5–23.9 (m), 22.6–22.1 (m), 22.3 (dd, *J* = 38.6, 18.5 Hz), and 21.4. HRMS (ESI, *m/z*): Calculated for $C_{11}H_9D_6NO$ [M + H]$^+$ 184.1603, found 184.1604.

(**23**). A colorless liquid after purification by flash column chromatography (petroleum ether/ethyl acetate = 40/1), 30.3 mg, with yield of 83%. $^1$H NMR (400 MHz, CDCl$_3$): δ 7.57 (d, *J* = 7.6 Hz, 1H), 7.29–7.25 (m, 1H), 7.05 (td, *J* = 7.6, 1.2 Hz, 1H), 6.88 (d, *J* = 8.0 Hz, 1H), 3.85 (q, *J* = 7.2 Hz, 2H), and 1.29 (t, *J* = 7.2 Hz, 3H). $^{13}$C{$^1$H} NMR (100 MHz, CDCl$_3$): δ 167.5, 154.5, 141.1, 127.5, 123.8, 123.5, 122.9, 121.4, 107.6, 34.1, 24.7–24.1 (m), 22.7–22.1 (m), and 12.9. HRMS (ESI, *m/z*): Calculated for $C_{11}H_9D_6NO$ [M + H]$^+$ 184.1603, found 184.1605.

## 4. Conclusions

In summary, we have developed a Fe(II) salt/hydrogen peroxide-promoted oxidative condensation of indoles with acetone or acetone-*d$_6$*, which offers an available and low-cost protocol for the synthesis of 3-alkylideneindolin-2-ones or 3-deuteroalkylideneindolin-2-ones with good yields. This approach shows a broad substrate scope and mild reaction conditions. In addition, this reaction system could be scaled up to 10 mmol easily. The oxidative alkenylation of other heterocycles is ongoing in our laboratory.

**Supplementary Materials:** The following supporting information can be downloaded at https://www.mdpi.com/article/10.3390/catal13060930/s1. Fail examples for the reaction system, characterization data for all products, X-ray diffraction analysis of compound **19**, $^1$H and $^{13}$C NMR spectra of all products, Deposition Numbers 2258769 (for **19**) contain the supplementary crystallographic data for this paper. These data are provided free of charge by the joint Cambridge Crystallographic Data Centre and Fachinformationszentrum Karlsruhe http://www.ccdc.cam.ac.uk/structures accessed on 4 May 2023. Numbers 2258769 (for 19) contain the supplementary crystallographic data for this paper [38].

**Author Contributions:** Conceptualization, Y.T.; methodology, Z.L. and Y.T.; investigation, Y.T., L.Z., Y.C., Y.L. and M.W.; writing—original draft preparation, Y.T.; writing—review and editing, Y.T. and Z.L.; supervision, W.F.; funding acquisition, Y.T. and Z.L. All authors have read and agreed to the published version of the manuscript.

**Funding:** The authors thank the Natural Science Foundation of Henan Province (202300410290) for financial support. We also thank the National Natural Science Foundation of China (21702044) and the Natural Science Foundation of Hebei Province (B2020201014, B2022201059) for their support.

**Data Availability Statement:** All experimental data are contained in the article and Supplementary Materials.

**Conflicts of Interest:** The authors declare no conflict of interest.

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
