# Peer review of "Ferrous Salt-Catalyzed Oxidative Alkenylation of Indoles: Facile Access to 3-Alkylideneindolin-2-Ones"

_catalysts, doi:10.3390/catal13060930_

Round 1
Reviewer 1 Report
A straightforward synthesis of 3-isopropylideneindolin-2-ones starting from an indole and acetone is described in this paper. This oxidative condensation is catalyzed by FeCl2 in the presence of H2O2. Optimal reaction conditions were found working with 5 mol% of the iron salt and 2 equiv of hydrogen peroxide, in a mixture of acetone and water (2.5/1, v/v), at 35 to 45 ºC, for 14 h. The present study could be considered an extension of previous contributions of the same research group in the synthesis of heterocycles involving radical processes. The resulting oxindole derivatives are of interest in pharmaceutical and material sciences. On the other hand, starting materials are readily available, and these reactions work well even at a gram-scale. Yields are from moderate to good for N-alkyl indoles. However, they are low for N-H derivatives (compounds 18-21), and it represents a weak point of this methodology.
In order to explain the oxidative, authors proposed a plausible speculative involving radical, anion and cation hydroxyl 2-hydroxy-2,3-dihydroindole derived intermediates.
The manuscript is written correctly, the products are adequately characterized, and the experimental work is well described in the Materials and Methods section.
I found this study of interest for synthetic organic chemists and therefore recommend it publication in Catalysts.
Minor remarks:
1. All the examples collected in this publication are with acetone and acetone-d6 as carbonyl compounds. Why are there no examples using other ketones? It seems that acetone is both a reagent and a solvent since a large excess of it is used.
2. Compounds 18-21 are solid products. Authors should provide the melting points of these compounds in the "Materials and Methods" section.
3. Authors should correct “5 mmol%” throughout the document, with “5 mol%” being the correct way to indicate the proportion of the catalyst.
Author Response
Question 1:All the examples collected in this publication are with acetone and acetone-d6 as carbonyl compounds. Why are there no examples using other ketones? It seems that acetone is both a reagent and a solvent since a large excess of it is used.
Response: Thanks for the comments. We have tried other ketones, such as 2-pentanone, 3-pentanone, 3-heptanone, 4-heptanone, cyclobutanone, cyclohexanone, and acetophenone before, but the corresponding products were not obtained under typical conditions, and most of starting materials were detected by TLC.
Question 2: Compounds 18-21 are solid products. Authors should provide the melting points of these compounds in the "Materials and Methods" section.
Response: Thanks for the comments. The melting points of compounds 18-21 have been added to the "Materials and Methods" section.
Question 3:Authors should correct “5 mmol%” throughout the document, with “5 mol%” being the correct way to indicate the proportion of the catalyst. Response: Thanks for the comments. The error has been corrected.
Reviewer 2 Report
The manuscript by Tian is devoted to the synthesis of the 3-alkylideneindolin-2-ones by oxidative coupling of ketones with indoles. The relevance and novelty of the article is beyond doubt, since indoles are a privileged scaffold in medicinal chemistry. New methods for the synthesis and functionalization of indoles are an important contribution.
The current manuscript and supporting information are prepared well. The main text requires some English language polishing. The reaction scope is quite nice, but no any other ketones than acetone are shown. The products are well characterized, but lacking mp’s for crystalline compounds. Please revise isotopic splitting in the 13C NMR for deuterated compounds.
In my opinion, the present work might be suitable to be published in Catalysts after minor revision
The main text requires some English language polishing.
Author Response
Question 1: The main text requires some English language polishing.
Response: Thanks for the comments. We have checked and revised the statements and grammar of the manuscript carefully.
Question 2:The reaction scope is quite nice, but no any other ketones than acetone are shown.
Response: Many thanks for the comments. Other ketones, such as 2-pentanone, 3-pentanone, 3-heptanone, 4-heptanone, cyclobutanone, cyclohexanone, and acetophenone was examined before. However, we failed to observe the desirable products, and most of starting materials were found by TLC.
Question 3:The products are well characterized, but lacking mp’s for crystalline compounds.
Response: Thanks for the comments. The melting points of solid compounds were added to the "Materials and Methods" section of the revised manuscript.
Question 4:Please revise isotopic splitting in the 13C NMR for deuterated compounds
Response: Many thanks for the helpful comments. We have revised the corresponding 13C NMR in revised manuscript.